# ZERO-SHOT RECOGNITION THROUGH IMAGE-GUIDED SEMANTIC CLASSIFICATION

## ABSTRACT

We present a new visual-semantic embedding method for generalized zero-shot learning. Existing embedding-based methods aim to learn the correspondence between an image classifier (visual representation) and its class prototype (semantic representation) for each class. Inspired by the binary relevance method for multi-label classification, we learn the mapping between an image and its semantic classifier. Given an input image, the proposed Image-Guided Semantic Classification (IGSC) method creates a label classifier, being applied to all label embeddings to determine whether a label belongs to the input image. Therefore, a semantic classifier is image conditioned and is generated during inference. We also show that IGSC is a unifying framework for two state-of-the-art deep-embedding methods. We validate our approach with four standard benchmark datasets.

## 1 INTRODUCTION

As a feasible solution for addressing the limitations of supervised classification methods, zero-shot learning (ZSL) aims to recognize objects whose instances have not been seen during training (Larochelle et al., 2008; Palatucci et al., 2009). Unseen classes are recognized by associating seen and unseen classes through some form of *semantic space*; therefore, the knowledge learned from seen classes is transferred to unseen classes. In the semantic space, each class has a corresponding vector representation called a *class prototype*. Class prototypes can be obtained from human-annotated attributes that describe visual properties of objects (Farhadi et al., 2009; Lampert et al., 2014) or from word embeddings learned in an unsupervised manner from text corpus (Mikolov et al., 2013; Pennington et al., 2014; Devlin et al., 2018).

A majority of ZSL methods can be viewed using the visual-semantic embedding framework, as displayed in Figure 1 (a). Images are mapped from the visual space to the semantic space in which all classes reside, or images and labels are projected to a latent space (Yang & Hospedales, 2015; Liu et al., 2018). Then, the inference is performed in this common space (Akata et al., 2013; Frome et al., 2013; Socher et al., 2013), typically using cosine similarity or Euclidean distance. Another perspective of embedding-based methods is to construct an image classifier for each unseen class by learning the correspondence between a binary one-versus-rest image classifier (i.e., visual representation of a class) and its class prototype in the semantic space (i.e., semantic representation of a class) (Wang et al., 2019). Once this correspondence function is learned, a binary one-versus-rest image classifier can be constructed for an unseen class with its prototype (Wang et al., 2019). For example, a commonly used choice for such correspondence is the bilinear function (Frome et al., 2013; Akata et al., 2013; 2015; Romera-Paredes & Torr, 2015; Li et al., 2018). Considerable efforts have been made to extend the linear function to nonlinear ones (Xian et al., 2016; Wang et al., 2017; Elhoseiny et al., 2017; Qiao et al., 2016). Figure 1 (b) illustrates this perspective.

Learning the correspondence between an image classifier and a class prototype has the following drawbacks. First, the assumption of using a single image classifier for each class is restrictive because the manner for separating classes in both visual and semantic spaces would not be unique. We argue that semantic classification should be conducted dynamically conditioned on an input image. For example, the visual attribute *wheel* may be useful for classifying most car images. Nevertheless, cars with missing wheels should also be correctly recognized using other visual attributes. Therefore, instance-specific semantic classifiers are more preferable than category-specific ones because the classifier weights can be adaptively determined based on image content. Second, the scale of

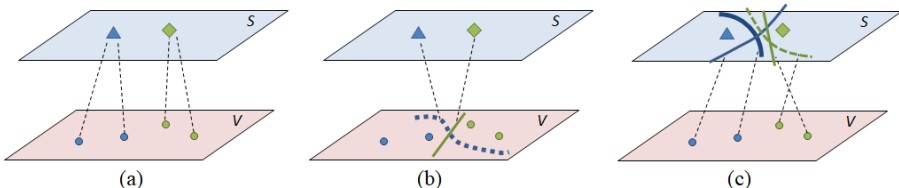

Figure 1: Zero-shot learning paradigms. (a) Conventional visual-to-semantic mapping trained on classification loss. (b) Another interpretation of visual-to-semantic mapping between visual and semantic representations. (c) The proposed IGSC, aiming to learn the correspondence between an image and a semantic classifier.

training data for learning the correspondence is constrained to be the number of class labels. In other words, a training set with $C$ labels has only $C$ visual-semantic classifier pairs to build the correspondence. This may hinder the robustness of deep models that usually require large-scale training data. Finally, although class embedding has rich semantic meanings, each class is represented by only a single class prototype to determine where images of that class collapse inevitably (MarcoBaroni, 2016; Fu et al., 2015). The mapped semantic representations from images may collapse to hubs, which are close to many other points in the semantic space, rather than being similar to the true class label (MarcoBaroni, 2016).

In this paper, we present a new method, named Image-Guided Semantic Classification (IGSC), to address these problems. IGSC aims to learn the correspondence between an image and its corresponding label classifier, as illustrated in Figure 1 (c). In contrast to existing methods focusing on the learning of visual (or semantic) representations (Zhang et al., 2016; Frome et al., 2013; Socher et al., 2013), IGSC analyzes the input image and seeks for combinations of variables in the semantic space (e.g., combinations of attributes) that distinguish a class (belonging to the input) from other classes. The proposed IGSC method has the following characteristics:

- IGSC learns the correspondence between an image in the visual space and a classifier in the semantic space. The correspondence can be learned with training pairs in the scale of training images rather than that of classes.

- IGSC performs learning to learn in an end-to-end manner. Label classification is conducted by a semantic classifier whose weights are generated on the fly. This model is simple yet powerful because of its adaptive nature.

- IGSC unifies visual attribute detection and label classification. This is achieved via the design of a conditional network (the proposed classifier learning method), in which label classification is the main task of interest and the conditional input image provides additional information of a specific situation.

- IGSC alleviates the hubness problem. The correspondence between an image and a semantic classifier learned from seen classes can be transferred to recognize unseen concepts.

We evaluated IGSC with experiments conducted on four public benchmark datasets, including SUN (Patterson & Hays, 2012), CUB (Patterson & Hays, 2012), AWA2 (Lampert et al., 2014), and aPY (Farhadi et al., 2009). Experimental results demonstrated that the proposed method achieved promising performance, compared with current state-of-the-art methods.The remainder of the paper is organized as follows: We briefly review related work in Section 2. Section 3 presents the proposed framework. The experimental results and conclusions are provided in Sections 4 and 5, respectively.

## 2    RELATED WORK

Zero-shot learning has evolved rapidly during the last decade, and therefore documenting the extensive literature with limited pages is rarely possible. In this section, we review a few representative zero-shot learning methods and refer readers to (Xian et al., 2019a; Wang et al., 2019) for a comprehensive survey. One pioneering main stream of ZSL uses attributes to infer the label of an image belonging to one of the unseen classes (Lampert et al., 2014; Al-Halah et al., 2016; Norouzi et al.,

2014; Jayaraman & Grauman, 2014; Kankuekul et al., 2012). The attributes of an image are predicted, then the class label is inferred by searching the class which attains the most similar set of attributes. For example, the Direct Attribute Prediction (DAP) model (Lampert et al., 2009) estimates the posterior of each attribute for an image by learning probabilistic attribute classifiers. A test sample is then classified by each attribute classifier alternately, and the class label is predicted by probabilistic estimation. Similar to the attribute-based methods, the proposed method has the merits of modeling the relationships among classes. However, IGSC unifies these two steps: attribute classifier learning and inferring from detected attributes to the class. Furthermore, attribute classifiers are jointly learned in IGSC.

A broad family of ZSL methods apply an embedding framework that directly learns a mapping from the visual space to the semantic space (Palatucci et al., 2009; Akata et al., 2013; 2015; Romera-Paredes & Torr, 2015). The visual-to-semantic mapping can be linear (Frome et al., 2013) or non-linear (Socher et al., 2013). For example, DeViSE (Frome et al., 2013) learns a linear mapping between the image and semantic spaces using an efficient ranking loss formulation. Cross-Modal Transfer (CMT) (Socher et al., 2013) uses a neural network with two hidden layers to learn a nonlinear projection from image feature space to word vector space. More recently, deep neural network models are proposed to mirror learned semantic relations among classes in the visual domain from the image (Annadani & Biswas, 2018) or from the part (Zhu et al., 2018a) levels. IGSC is also an embedding-based ZSL method. IGSC differs significantly from existing methods in that IGSC learns the correspondence between an image and its semantic classifier, enabling the possibility of using different classification manners to separate class prototypes in the semantic space.

Recent ZSL models adopt the generative adversarial network (GAN) (Goodfellow et al., 2014) or other generative models for synthesizing unseen examples (Bucher et al., 2017; Long et al., 2017; Jiang et al., 2018; Verma et al., 2018; Xian et al., 2018; Zhu et al., 2018b; Xian et al., 2019b; Verma et al., 2020; Yu et al., 2020; Ma & Hu, 2020) or for reconstructing training images (Chen et al., 2018). The synthesized images obtained at the training stage can be fed to conventional classifiers so that ZSL is converted into the conventional supervised learning problem (Long et al., 2017). The transformation from attributes to image features require involving generative models such as denoising autoencoders (Bucher et al., 2017), GAN (Xian et al., 2018; Zhu et al., 2018b) or their variants (Verma et al., 2018; Felix et al., 2018; Xian et al., 2019b; Yu et al., 2020; Ma & Hu, 2020). Despite outstanding performances reported in the papers, these works leverage some form of the unseen class information during training. In view of real-world applications involving recognition in-the-wild, novel classes including the image samples as well as the semantic representations may not be available in model learning. The proposed method is agnostic to all unseen class information during training. Furthermore, the proposed method is much simpler in the architecture design and has a much smaller model size, compared with the generative methods.

## 3 APPROACH

### 3.1 PROBLEM DESCRIPTION

Given a training set $S = \{(x_n, y_n), n = 1 \ldots N\}$, with $y_n \in \mathcal{Y}_s$ being a class label in the seen class set, the goal of ZSL is to learn a classifier $f : \mathcal{X} \to \mathcal{Y}$ which can generalize to predict any image $x$ to its correct label, which is not only in $\mathcal{Y}_s$ but also in the unseen class set $\mathcal{Y}_u$. In the prevailing family of compatibility learning ZSL (Xian et al., 2019a; Ba et al., 2015), the prediction is made via:

$$\hat{y} = f(x; W) = \arg\max_{y \in \mathcal{Y}} F(x, y; W). \tag{1}$$

In particular, if $\mathcal{Y} = \mathcal{Y}_u$, this is the conventional ZSL setting; if $\mathcal{Y} = \mathcal{Y}_s \cup \mathcal{Y}_u$, this is the generalized zero-shot learning (GZSL) setting, which is more practical for real-world applications. The compatibility function $F(\cdot)$—parameterized by $W$—is used to associate visual and semantic information.

In the visual space, each image $x$ has a vector representation, denoted by $\theta(x)$. Similarly, each class label $y$ has a vector representation in the semantic space (called the class prototype), denoted by $\phi(y)$. In short, $\theta(x)$ and $\phi(y)$ are the image and class embeddings, both of which are given.

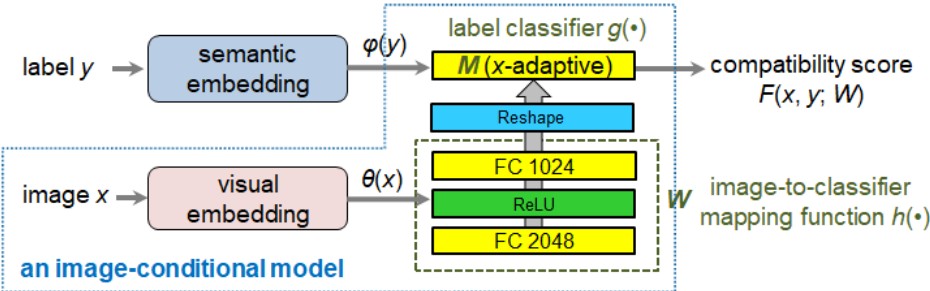

Figure 2: The architecture of IGSC. This model receives an image and a label, and it returns the compatibility score of this input pair. The score indicates the probability of the label belonging to the image. The score is calculated by a label classifier $g(\cdot)$, whose weights $M$ are stored in the output layer of a fully connected neural network. Therefore, weight values depend on the input image. The neural network is characterized by the parameters $W$, which are the only parameters required to learn from training data.

## 3.2 IMAGE-GUIDED SEMANTIC CLASSIFICATION MODEL

The compatibility function in this work is achieved by implementing two functions, $h(\theta(x); W)$ and $g(\phi(y); M)$, as illustrated in Figure 2. The first function $h(\cdot)$ receives an image embedding as input and returns parameters $M$ characterizing a label classifier:

$$M = h(\theta(x); W). \tag{2}$$

In other words, $h(\cdot)$ learns the mapping between image representations and model (i.e., semantic classifier) representations. Each image has its own semantic classifier. Images of the same class may have different classifier weights. Different from existing methods where the classifier weights are part of model parameters and thereby being static after training, the classifier weights in IGSC are dynamically generated during test time.

The image-to-classifier mapping can be either linear or nonlinear. Figure 2 shows an implementation of a nonlinear model that involves two fully connected layers and an output layer. The dimension of the output layer is set to accommodate the label classifier weights. We emphasize again that $W$ are the only model parameters required to learn from training data.

The second function $g(\cdot)$ is a label classifier, characterized by the parameters outputted by $h(\cdot)$. This function takes a label vector as input, and returns a prediction score indicating the probability of the label belonging to the input image:

$$s = g(\phi(y); M). \tag{3}$$

Let $s_j$ denote the prediction score for a label $j$. In multi-class (single-label) image classification, the final compatibility score is obtained by normalizing the prediction scores to probabilistic values with softmax:

$$F(x, y_j; W) = \frac{\exp(s_j)}{\sum_{k=1}^{|\mathcal{Y}|} \exp(s_k)}. \tag{4}$$

One image is assigned to the class with the highest compatibility score. In multi-label image classification, we replace softmax by a sigmoid activation function. The prediction is made by choosing labels whose compatibility score is greater than a threshold.

It is worth noting that the mechanism of IGSC is similar to that of *Dynamic Filter Networks* (Jia et al., 2016), in which the filters are generated dynamically conditioned on an input. A similar mechanism also appears in (Zhao et al., 2018), which predicts a set of adaptive weights from conditional inputs to linearly combine the basis filters. The proposed method differs fundamentally in that both (Jia et al., 2016) and (Zhao et al., 2018) focus on learning image representations, while our method aims to learn model representations that are applied to a different modality (i.e., labels).

### 3.3 FORMS OF LABEL CLASSIFIERS

The image-guided label classifier can be either linear or nonlinear, which receives a label embedding and returns a prediction score of the label. In this study we experiment with two variations of the label classifier. The linear label classifier is represented as:

$$g(\phi(y); M) = \mathbf{m}\phi(y) + b. \tag{5}$$

where $\mathbf{m} \in \mathbb{R}^d$ is a weight vector, $b$ is a threshold and $M = (\mathbf{m}, b)$. The dimension $d$ is set to that of the label vector (e.g., $d = 300$ if using 300-dim word2vec (Mikolov et al., 2013)). Alternatively, the nonlinear label classifier is implemented using a two-layer neural network:

$$g(\phi(y); M) = \mathbf{m_2} \tanh(\mathbf{M_1}\phi(y) + b_1) + b_2, \tag{6}$$

where $\mathbf{M_1} \in \mathbb{R}^{h \times d}, \mathbf{m_2} \in \mathbb{R}^h$ and $M = (\mathbf{M_1}, b_1, \mathbf{m_2}, b_2)$. The nonlinear classifier characterizes the $d$-dim semantic space by using $h$ perceptrons and performs the classification task. We set $h$ to 30 in the experiments. As will be shown in Section 4, the nonlinear label classifier outperforms a linear one.

For GZSL, it is beneficial to enable *calibrated stacking* (Chao et al., 2016), which reduces the scores for seen classes. This leads to the following modification:

$$\hat{y} = \arg\max_{y \in \mathcal{Y}_s \cup \mathcal{Y}_u} \big(g(\phi(y); M) - \gamma \mathbb{1}[y \in \mathcal{Y}_s]\big), \tag{7}$$

where $\mathbb{1}[y \in \mathcal{Y}_s] \in \{0, 1\}$ indicates whether or not $y$ is a seen class and $\gamma$ is a calibration factor.

### 3.4 LEARNING MODEL PARAMETERS

Recall that the objective of ZSL is to correctly assign the correct label to an image. This is a typical classification problem. For a training sample $x_i$, Let $y_i = \{y_i^1, y_i^2, ..., y_i^{|\mathcal{Y}_s|}\} \in \{0, 1\}$ denote the one-hot encoding of the ground truth label and $p_i = \{p_i^1, p_i^2, ..., p_i^{|\mathcal{Y}_s|}\}$ denote the compatibility scores of $x_i$ (Equ. 4). That is, $p_i^j = F(x_i, y_j; W)$. The model parameters $W$ are learned by minimizing the cross entropy loss:

$$\mathcal{L} = -\sum_{i=1}^{N} \sum_{j=1}^{|\mathcal{Y}_s|} y_i^j \log(p_i^j) + (1 - y_i^j)\log(1 - p_i^j). \tag{8}$$

We had also experimented with the hinge loss and achieved similar performances. The model parameters including $W$ and those of the image/semantic embedding networks can be jointly learned end-to-end; however, the results reported in Section 4 were obtained by freezing the weights of feature extractors for a fair comparison. That is, all methods under comparison used the same image and semantic representations in the experiments.

### 3.5 CONNECTION TO PREVIOUS METHODS

Finally we show how previous supervised visual-semantic embedding methods—DeViSE (Frome et al., 2013) and CMT (Socher et al., 2013)—are special cases of our method.

DeViSE (Frome et al., 2013) uses a projection layer (a linear transformation) that maps a visual vector to the semantic space, and compute a dot-product similarity between the projected visual vector and the vector representation of the correct label. The behavior is identical to a special case of our method, where both $h(\cdot)$ and $g(\cdot)$ are linear. CMT (Socher et al., 2013) uses a neural network with two hidden layers and the standard nonlinearity $tanh$ to learn a nonlinear projection from image feature space to word vector space and compute the Euclidean distances of the L2 normed vectors. This is identical to the special case of using nonlinear $h(\cdot)$ and linear $g(\cdot)$, except that we use ReLU instead of $tanh$ in the nonlinear transformation.

## 4 EXPERIMENTS

### 4.1 DATASETS AND EXPERIMENTAL SETTING

We used four popular benchmark datasets, including coarse-grained and fine-grained datasets, for evaluating the proposed method. The statistics of the datasets are summarized in Table 1. Please see

Table 1: Summary of the datasets used in the experiments

| Dataset | Embedding dim. | Number of classes | | Number of samples | | | |
|---|---|---|---|---|---|---|---|
| | | Seen | Unseen | Training | Test (seen) | Test (unseen) | Total |
| SUN (Patterson & Hays, 2012) | 102 | 580 + 65 | 72 | 10,320 | 2,580 | 1,440 | 14,340 |
| CUB (Welinder et al., 2010) | 312 | 100 + 50 | 50 | 7,057 | 2,967 | 1,764 | 11,788 |
| AWA2 (Lampert et al., 2014) | 85 | 27 + 13 | 10 | 23,527 | 5,882 | 7,913 | 37,322 |
| aPY (Farhadi et al., 2009) | 64 | 15 + 5 | 12 | 5,932 | 1,483 | 7,924 | 15,339 |

(Xian et al., 2019a) for detailed descriptions. We followed the new split provided by (Xian et al., 2019a) because this split ensured that classes at test should be strictly unseen at training.

**Visual and semantic embeddings.** For a fair comparison, we used the 2048-dimensional ResNet-101 features provided by (Xian et al., 2019a) as image representations. For label representations, we used the semantic embeddings provided by (Xian et al., 2019a), each of which is an L2-normalized attribute vector. Note that IGSC is flexible in that the visual and semantic embeddings, $h(\cdot)$ and $g(\cdot)$ functions can all be customized to meet specific needs.

**Training details.** We used Adaptive Moment Estimation (Adam) for optimizing the model. We augmented the data by random cropping and mirroring. The learning rate was set fixed to $10^{-5}$. Training time for a single epoch ranged from 91 seconds to 595 seconds (depending on which dataset was used). Training the models using four benchmark datasets roughly took 11 hours in total. The runtime was reported running on a machine with an Intel Core i7-7700 3.6-GHz CPU, NVIDIA's GeForce GTX 1080Ti and 32 GB of RAM. The dimension in the nonlinear variant of the semantic classifier $g(\cdot)$ was set to 30 in the experiments.

**Evaluation protocols.** We followed the standard evaluation metrics used in the literature. For ZSL, we used average per-class top-1 accuracy as the evaluation metric, where the prediction (Eq. 1) is successful if the predicted class is the correct ground truth. For GZSL, we reported $acc_s$ (test images are from seen classes and the prediction labels are the union of seen and unseen classes) and $acc_u$ (test images are from unseen classes and the prediction labels are the union of seen and unseen classes). We computed the harmonic mean (Xian et al., 2019a) of accuracy rates on seen classes $acc_s$ and unseen classes $acc_u$:

$$H = \frac{2 \times acc_s \times acc_u}{acc_s + acc_u}. \qquad (9)$$

The harmonic mean offers a comprehensive metric in evaluating GZSL methods. The harmonic mean value is high only when both accuracy rates are high. We reported the average results of three random trials for each ZSL and GZSL experiment.

Table 2: Ablation study on effects of different design choices

| | ZSL | GZSL | | |
|---|---|---|---|---|
| | acc | $acc_u$ | $acc_s$ | H |
| linear $h$ + linear $g$ | 56.97 | 19.56 | 28.71 | 23.27 |
| linear $h$ + nonlinear $g$ | 54.56 | 17.15 | 31.99 | 22.32 |
| nonlinear $h$ + linear $g$ | 58.01 | 19.68 | 31.08 | 24.10 |
| nonlinear $h$ + nonlinear $g$ | 58.30 | 19.88 | 36.41 | 25.72 |

## 4.2 ABLATION STUDY

First, we investigate the effects of different designs of the image-to-classifier mapping function $h(\cdot)$ and the label classifier $g(\cdot)$. We reported the results on the SUN benchmark; however, similar findings can be found using other datasets.

Table 2 shows the results of the ablation experiment. In both settings (ZSL and GZSL), using a *nonlinear* image-to-classifier mapping (i.e., $h(\cdot)$) is essential to the performance. A significant performance gain was observed when a nonlinear $h(\cdot)$ was applied. The combination of linear $h(\cdot)$ and nonlinear $g(\cdot)$ performed the worst. A possible reason is that a linear mapping does not have a sufficient capacity to model the relation between a visual feature and its corresponding semantic

Table 3: Standard zero-shot learning results (top-1 accuracy) on four benchmark datasets

| Method | SUN | CUB | AWA2 | aPY |
|---|---|---|---|---|
| DAP (Lampert et al., 2009) | 39.9 | 40.0 | 46.1 | 33.8 |
| IAP (Lampert et al., 2009) | 19.4 | 24.0 | 35.9 | 36.6 |
| CONSE (Norouzi et al., 2014) | 38.8 | 34.3 | 44.5 | 26.9 |
| CMT (Socher et al., 2013) | 39.9 | 34.6 | 37.9 | 28.0 |
| SSE (Zhang & Saligrama, 2015) | 51.5 | 43.9 | 61.0 | 34.0 |
| LATEM (Xian et al., 2016) | 55.3 | 49.3 | 55.8 | 35.2 |
| ALE (Akata et al., 2013) | 58.1 | 54.9 | 62.5 | 39.7 |
| DeViSE (Frome et al., 2013) | 56.5 | 52.0 | 59.7 | **39.8** |
| SJE (Akata et al., 2015) | 53.7 | 53.9 | 61.9 | 32.9 |
| ESZSL (Romera-Paredes & Torr, 2015) | 54.5 | 53.9 | 58.6 | 38.3 |
| SYNC (Changpinyo et al., 2016) | 56.3 | 55.6 | 46.6 | 23.9 |
| SAE (Kodirov et al., 2017) | 40.3 | 33.3 | 54.1 | 8.3 |
| GFZSL (Verma & Rai, 2017) | **60.6** | 49.3 | **63.8** | 38.4 |
| IGSC | 58.3 | **56.9** | 62.1 | 35.2 |

Table 4: Generalized zero-shot learning results (top-1 accuracy and H) on four benchmark datasets. All methods are agnostic to both unseen images and unseen semantic vectors during training.

| Method | SUN | | | CUB | | | AWA2 | | | aPY | | |
|---|---|---|---|---|---|---|---|---|---|---|---|---|
| | $acc_u$ | $acc_s$ | H | $acc_u$ | $acc_s$ | H | $acc_u$ | $acc_s$ | H | $acc_u$ | $acc_s$ | H |
| DAP(Lampert et al., 2009) | 4.2 | 25.1 | 7.2 | 1.7 | 67.9 | 3.3 | 0.0 | 84.7 | 0.0 | 4.8 | 78.3 | 9.0 |
| IAP(Lampert et al., 2009) | 1.0 | 37.8 | 1.8 | 0.2 | 72.8 | 0.4 | 0.9 | 87.6 | 1.8 | 5.7 | 65.6 | 10.4 |
| CONSE(Norouzi et al., 2014) | 6.8 | 39.9 | 11.6 | 1.6 | 72.2 | 3.1 | 0.5 | **90.6** | 1.0 | 0.0 | **91.2** | 0.0 |
| CMT(Socher et al., 2013) | 8.1 | 21.8 | 11.8 | 7.2 | 49.8 | 12.6 | 0.5 | 90.0 | 1.0 | 1.4 | 85.2 | 2.8 |
| CMT*(Socher et al., 2013) | 8.7 | 28.0 | 13.3 | 4.7 | 60.1 | 8.7 | 8.7 | 89.0 | 15.9 | 10.9 | 74.2 | 19.0 |
| SSE(Zhang & Saligrama, 2015) | 2.1 | 36.4 | 4.0 | 8.5 | 46.9 | 14.4 | 8.1 | 82.5 | 14.8 | 0.3 | 78.9 | 0.4 |
| LATEM(Xian et al., 2016) | 14.7 | 28.8 | 19.5 | 15.2 | 57.3 | 24.0 | 11.5 | 77.3 | 20.0 | 0.1 | 73.0 | 0.2 |
| ALE(Akata et al., 2013) | 21.8 | 33.1 | 26.3 | 23.7 | 62.8 | 34.4 | 14.0 | 81.8 | 23.9 | 4.6 | 73.7 | 8.7 |
| DEVISE(Frome et al., 2013) | 16.9 | 27.4 | 20.9 | 23.8 | 53.0 | 32.8 | 17.1 | 74.7 | 27.8 | 4.9 | 76.9 | 9.2 |
| SJE(Akata et al., 2015) | 14.7 | 30.5 | 19.8 | 23.5 | 59.2 | 33.6 | 8.0 | 73.9 | 14.4 | 3.7 | 55.7 | 6.9 |
| ESZSL(Romera-Paredes & Torr, 2015) | 11.0 | 27.9 | 15.8 | 12.6 | 63.8 | 21.0 | 5.9 | 77.8 | 11.0 | 2.4 | 70.1 | 4.6 |
| SYNC(Changpinyo et al., 2016) | 7.9 | **43.3** | 13.4 | 11.5 | 70.9 | 19.8 | 10.0 | 90.5 | 18.0 | 7.4 | 66.3 | 13.3 |
| SAE(Kodirov et al., 2017) | 8.8 | 18.0 | 11.8 | 7.8 | 54.0 | 13.6 | 1.1 | 82.2 | 2.2 | 0.4 | 80.9 | 0.9 |
| GFZSL(Verma & Rai, 2017) | 0.0 | 39.6 | 0.0 | 0.0 | 45.7 | 0.0 | 2.5 | 80.1 | 4.8 | 0.0 | 83.3 | 0.0 |
| SP-AEN (Chen et al., 2018) | 24.9 | 38.6 | 30.3 | 34.7 | 70.6 | 46.6 | 23.3 | 90.9 | 37.1 | 13.7 | 63.4 | 22.6 |
| PSR(Annadani & Biswas, 2018) | 20.8 | 37.2 | 26.7 | 24.6 | 54.3 | 33.9 | 20.7 | 73.8 | 32.3 | 13.5 | 51.4 | 21.4 |
| AREN (Xie et al., 2019) | 19.0 | 38.8 | 25.5 | 38.9 | **78.7** | **52.1** | 5.6 | 92.9 | 26.7 | 9.2 | 76.9 | 16.4 |
| IGSC | 19.8 | 36.4 | 25.7 | 27.8 | 66.8 | 39.3 | 19.8 | 84.9 | 32.1 | 13.4 | 69.5 | 22.5 |
| IGSC+CS | **39.4** | 31.3 | **34.9** | 40.8 | 60.2 | 48.7 | **25.7** | 83.6 | **39.3** | **23.1** | 58.9 | **33.2** |

classifier, and using a nonlinear $g(\cdot)$ exacerbates the overfitting problem of learned semantic classifiers to seen classes. While a nonlinear $h(\cdot)$ successfully modeled the mapping between a visual feature and its label classifier, using a nonlinear $g(\cdot)$ further improved the recognition performance, especially under the setting of GZSL.

## 4.3 COMPARISONS WITH STATE-OF-THE-ART EMBEDDING-BASED APPROACHES

We compared the IGSC method with a variety of standard and generalized ZSL methods as reported in (Xian et al., 2019a). These methods can be categorized into 1) attribute-based: DAP (Lampert et al., 2009), IAP (Lampert et al., 2009), CONSE (Norouzi et al., 2014), SSE (Zhang & Saligrama, 2015), SYNC (Changpinyo et al., 2016); and 2) embedding-based: CMT/CMT* (Socher et al., 2013), LATEM(Xian et al., 2016), ALE(Akata et al., 2013), DeViSE(Frome et al., 2013), SJE(Akata et al., 2015), ESZSL(Romera-Paredes & Torr, 2015), SAE(Kodirov et al., 2017), GFZSL(Verma & Rai, 2017). Performances of the methods are directly reported from the paper (Xian et al., 2019a).

Please note that all methods under comparison—including the proposed method—are *inductive to both unseen images and unseen semantic vectors*. Only labeled training instances and class prototypes of seen classes are available in this experimental setting. Alternatively, methods that are transductive for unseen class prototypes and unlabeled unseen test instances can achieve better performances because more information is involved in model learning. Recent methods in the inductive setting are only inductive to samples (Jiang et al., 2018; Felix et al., 2018; Xian et al., 2019b; Schonfeld et al., 2019; Verma et al., 2020; Yu et al., 2020; Ma & Hu, 2020; Huynh & Elhamifar, 2020).

Table 5: $N_1$ skewness on SUN benchmark.

| | ZSL | | GZSL | |
|---|---|---|---|---|
| | test (seen) | test (unseen) | test (seen) | test (unseen) |
| DeViSE (Frome et al., 2013) | 2.163 | 2.360 | 2.355 | 2.849 |
| IGSC | **1.046** | **0.380** | **0.111** | **2.452** |

These methods use unseen class labels during training, which is different to our setting and, therefore, are not compared.

We reported the performance the proposed IGSC method with (or without) calibrated stacking (Equ. 7): 1) **IGSC** uses the nonlinear-nonlinear combination; 2) **IGSC+CS** enables calibrated stacking. Table 3 shows the conventional ZSL results. IGSC has a superior performance to those of other methods on the CUB dataset and achieves comparable performances on the other datasets. Although GFZSL (Verma & Rai, 2017) has the best performances on SUN and AWA2, this method performs poorly under the GZSL setting.

Table 4 shows the generalized ZSL results. In this experiment, recent inductive methods (Chen et al., 2018; Annadani & Biswas, 2018; Xie et al., 2019) are included for comparison. The semantics-preserving adversarial embedding network (SP-AEN) (Chen et al., 2018) is a GAN-based method, which uses an adversarial objective to reconstruct images from semantic embeddings. The preserving semantic relations (PSR) method (Annadani & Biswas, 2018) is an embedding-based approach utilizing the structure of the attribute space using a set of relations. Finally, the attentive region embedding network (AREN) (Xie et al., 2019) uses an attention mechanism to construct the embeddings from the part level (i.e., local regions), which consists of two embedding streams to extract image regions for semantic transfer.

By examining the harmonic mean values, IGSC consistently outperformed other competitive methods on three out of the four datasets. The performance gain validated the effectiveness of learning image-guided semantic classifiers. Compared with embedding based methods, this novel paradigm not only has more training pairs (in the scale of the training images) for learning the correspondence between an image and its corresponding label classifier but also allows different manners to separate classes based on the content of input image. In comparison with attribute based methods which take a two-step pipeline to detect attributes from one image and aggregate the detection results for label prediction, IGSC unifies the steps. Compared with recent methods (Chen et al., 2018; Annadani & Biswas, 2018; Xie et al., 2019), IGSC is much simpler and therefore has a greater flexibility. We have not integrated powerful components for GZSL such as feature generators and attention models, yet it has achieved comparable (or superior) performance to existing sophisticated methods.

An additional reason that explains the improved performance is that the hubness may be alleviated in IGSC, which avoids nearest neighbor searches of class prototypes in the semantic space. We conducted an experiment to realize whether IGSC reduces hubness. To measure the degree of hubness, we used the skewness of the empirical $N_1$ distribution (Radovanović et al., 2010; Shigeto et al., 2015). We conducted this experiment on the SUN benchmark because it is the only dataset containing an equal number of test images per class. As we hardly found the skewness analyses in recent literature, we implemented DeViSE (Frome et al., 2013) and compare it with the proposed method. The results are summarized in Table 5. IGSC produced smaller skewness values. One possible reason explaining why hubness is alleviated is that the "matching" between a visual representation and a class prototype is more flexible in IGSC than that in nearest neighbor search. A label is considered correct as long as its embedding is on the right side of the decision surface, learned conditioned on the input image embedding.

To better understand the strengths and weaknesses of the proposed method, we compare its performance with recent GAN-based methods. The notable performance of f-VAEGAN-D2 (Xian et al., 2019b) achieved the harmonic mean values of 41.3 on SUN and 53.6 on CUB. This GAN-based approach and its variants focus on the design of a data augmentation scheme where arbitrarily many synthetic features of both seen and unseen classes can be created to aid in improving the discriminative power of classifiers. Linear softmax classifiers are typically used in such approaches. Alternatively, the proposed embedding-based method focuses on the design of classifiers. These techniques may complement each other to advance methods for zero-shot learning.

Table 6: Ablation study on effects of different visual models

| Backbone | SUN | | | CUB | | | AWA2 | | | aPY | | |
|---|---|---|---|---|---|---|---|---|---|---|---|---|
| | $acc_u$ | $acc_s$ | $H$ | $acc_u$ | $acc_s$ | $H$ | $acc_u$ | $acc_s$ | $H$ | $acc_u$ | $acc_s$ | $H$ |
| Res-101 | 22.5 | 36.1 | 27.7 | 27.8 | 66.8 | 39.3 | 19.8 | 84.9 | 32.1 | 13.4 | 69.5 | 22.5 |
| Res-152 | 23.7 | 36.1 | 28.6 | 28.6 | 68.0 | 40.3 | 22.7 | 83.9 | 35.7 | 14.9 | 67.6 | 24.4 |
| Res-101+CS | 39.4 | 31.3 | 34.9 | 40.8 | 60.2 | 48.7 | 25.7 | 83.6 | 39.3 | 23.1 | 58.9 | 33.2 |
| Res-152+CS | 40.8 | 31.2 | 35.3 | 42.9 | 61.0 | 50.4 | 27.1 | 83.0 | 40.9 | 26.4 | 53.3 | 35.3 |

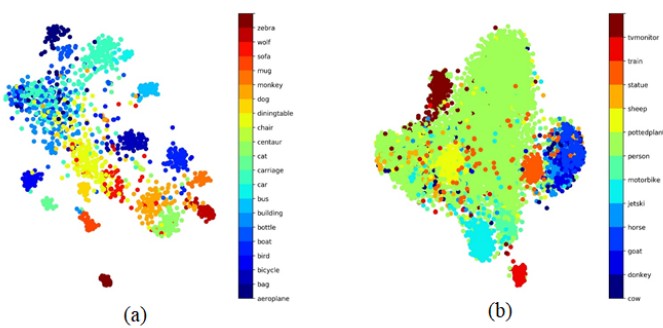

Figure 3: t-SNE visualization of the model space learned by IGSC: (a) seen classes, (b) unseen classes. Best viewed in color.

## 4.4 DISCUSSIONS

We discuss the model flexibility and visualize the classifier weights generated by IGSC.

**Model extendibility.** The proposed IGSC model is flexible in that the visual and semantic embeddings, the $h(\cdot)$ and $g(\cdot)$ functions can all be customized to meet specific needs. We provide a proof of concept analysis, in which we investigate the effect of replacing Res-101 with Res-152. Table 6 shows the result. Performance improvements are observed when we use a deeper visual model. By elaborating other components in IGSC, it seems reasonable to expect this approach should yield even better results.

**Model visualization.** Finally, we visualize the "model representations" of the label classifiers by using t-SNE (van der Maaten & Hinton, 2008) of the dynamically generated classifier weights. Figure 3 displays the visualization results using the aPY dataset. Each point in the figure represents a label classifier generated on-the-fly by feeding a test image to the IGSC model. Colors indicate class labels. Although each image has its own label classifier, the IGSC method tends to generate similar model representations for images of the same class (Fig. 3 (a)). Please note that this framework allows the learning of a "single" label classifier for each class (class-level classifier), i.e., the model representations for images of the same class are identical. However, the results show that instance-level classifiers benefit the optimization of recognition accuracy (points are scattered but fall around).

## 5 CONCLUSION

We propose a unifying visual-semantic embedding model that transform an image into a label classifier, consequently used to predict the correct label in the semantic space. Modeling the correspondence between an image and its label classifier enables a powerful GZSL method that achieves promising performances on four benchmark datasets. One future research direction we are pursuing is to extend the method for multi-label zero-shot learning, in which images are assigned with multiple labels from an open vocabulary. This would take full advantage of the semantic space. Another direction is to explore model learning with a less restricted setting, which can be transductive for specific unseen classes or test instances.

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
