# OpenReview forum: "Zero-Shot Recognition through Image-Guided Semantic Classification"
_ICLR.cc/2021/Conference — Reject_

### Official Review · AnonReviewer3 · 2020-10-27
**A simple yet effective method for zero-shot learning**

**Rating:** 4
**Confidence:** 5

**Review:**

------------
Summary: This paper proposes a simple yet effective method for zero-shot learning. In the method, a network is learned to predict the compatibility function weight given the input of the image. The predicted weight is then applied to semantic attributes and the final class label is predicted by the maximum compatibility score. The method is evaluated on benchmark datasets and illustrates competitive performance.

------------

Reason for score:
The paper is overall clear and easy to understand, and the proposed method is simple and achieves competitive results. But the paper lacks significance and contributions (detailed below), which makes me hard to judge its potential impact in the related area . So I don't find the paper meet the standard of ICLR. Hopefully the authors can address my concerns in the rebuttal period.

------------

Pros:
1. The authors propose a simple method for zero-shot learning motivated by learning instance-guided classifier weights.
2. The method demonstrates competitive results on benchmark datasets with a very simple implementation.
3. The paper is written clearly and easy to follow.

------------
Cons:
1. The paper fails to validate its motivation. The authors claim the method is proposed to learn an "image-guided semantic classification model" and the "weight values depend on the input image". Although the model architecture complies the motivation and the evaluation results are good, it is hard to tell if the results are achieved by the "image-guided weight values". The authors didn't provide either theoretical analysis or empirical evaluations to demonstrate that the weights learned by the model is image-sensitive and the overall performance benefits from it. For example, it will be helpful if the authors can visualize the
2. Some implementation details are missing and some choices in the method lack explanations. For example, why the model is trained by a cross-entropy loss in (8)? Is it specifically beneficial to the proposed method in the zero-shot learning context? For the non-linear label classifier, what is the hidden layer dimension $h$ used in the experiments? What is the effect of extending the non-linear label classifier to deeper networks? Will it saturate? etc. The paper lacks this type of details or evaluations, which makes the readers hard to evaluate its potential effectiveness.
3. The proposed method is a framework which can implemented in various ways, but the authors fail to provide enough analysis on its potential extensions. For example, in the experiments, for fair comparison the authors froze the feature extractor. But it will be helpful if the authors can also provide the performance with the feature extractor fine-tuned, so as to provide insight on how this framework will synergize with deep feature networks. In addition, I'd love to see the performance of the proposed method on large-scale recognition-in-the-wild dataset like ImageNet.
4. Although the authors fix the feature extractor and only compare with the methods in the exact same setting (namely, feature extractor fixed, no unseen information in training), the baselines compared are rather weak. For example, the baseline models may not benefit from the Calibrated Stacking (CS),  and the proposed method without CS has less significant results. Some baseline models like SP-AEN have lower unseen and higher seen than the proposed method, so it is possible they can achieve better performance with CS. Also, like we discussed in 3, many feature-based methods achieve significantly better results and it will be great to show how the proposed method works if the entire model including the feature extractor is trained end-to-end.

Based on the points above, I find the paper less significant and lack contributions, thus giving my score.

---

> ### Author Response · Authors · 2020-11-23
> **Responses to AnonReviewer3**
>
> 1. Thank you for your suggestion. In the paper we have added Section 4.4 and Fig. 3, involving a visualization of the weight values that validates the motivation of learning image-adaptive classification weights. Two interesting observations are: (1) our method tends to generate similar weight values for images of the same class; (2) Although the proposed framework allows the learning of a “single” label classifier for each class (class-level classifier), instance-level classifiers benefit the optimization of recognition accuracy (points are “scattered” but fall around).
>
> 2. The cross-entropy loss in (8) is a reasonable choice. It is not specifically beneficial to the proposed method. We had experimented with the log-sum-exp pairwise loss function (a smooth version of the hinge loss) and achieved a similar performance but with a longer training time. For the non-linear label classifier, the hidden layer dimension $h$ was set to 30 in all experiments. Conceptually the nonlinear classifier can be very deep, say, we can use a Res-101 to generate image-adaptive Res101s. We had added one more layer but did not find improved performance. However, we were not sure if the performance saturated because we experimented with the ZSL benchmark datasets (not very large).
>
> 3. We agree that we should provide the performance with the feature extractor fine-tuned, but it is sad to tell that the machine we can afford does not have sufficient memory to conduct this experiment. Instead, we had replaced the Res-101 features with Res-152 and showed the results in Section 4.4 and Table 6. Performance improvements were observed when we extracted visual features by using a deeper visual model.
>
> 4. Please refer to our 3rd response to AnonReviewer2.

---

### Official Review · AnonReviewer1 · 2020-10-27
**A reasonable ZSL formulation with weak experimental results**

**Rating:** 3
**Confidence:** 4

**Review:**

This paper describes zero-shot learning for image classification.  The proposed method is termed as Image-Guided Semantic Classification (IGSC) that learns image-specific label classifiers to achieve zero-shot classification. Experimental results on four standard datasets are reported.

The authors are suggested to address the following comments.
1. The idea to learn image-specific classifiers (specified by the yielded parameters) resembles the conventional local learning, where a classifier is learned for each training sample. In the last paragraph of page 4, the authors mention that the proposed IGSC mechanism is similar to that of  Dynamic Filter Networks by Jia et al., but  also point out a fundamental difference that their method instead aims to learn model representations. However, similar techniques in this aspect have already been explored, e.g., in solving the VQA problem. Take, for example, the CVPR 2016 paper, entitled “Image Question Answering using Convolutional Neural Network with Dynamic Parameter Prediction.” The novelty of the proposed image-guided formulation seems to be limited.
2. The math notations in the sentence right after equation (1) are incorrect. Y \in Y_u should be corrected into Y = Y_u. Also, Y in Y_s \cup Y_u should be Y = Y_s \cup Y_u.
3. Besides the technical novelty, my main concern about this work is that the experimental results are not convincing. For example, the previous technique AREN (Xie et al., 2019) is included in Table 4, but not in Table 3. However, the classification results achieved by AREN are considerably better than those yielded by the proposed technique.  For the ZSL results in Table 3, AREN achieves 60.6/71.8/67.9/39.2, while those by the proposed method are 58.3/56.9/62.1/35.2. For the GZSL results in Table 4, the acc_u result of SUN by AREN is 19%, not 9%. Furthermore, as Calibrated Stacking (CS) is also used in AREN, it is not reasonable that the comparison excludes the GZSL results of AREN+CS, which are significantly better than those by the proposed IGSC+CS. In particular, the respective harmonic means derived by AREN+CS are 35.9/66.0/64.7/36.9, while those by the proposed technique (IGSC+AC) are 34.9/48.7/39.3/33.2.
4. The comparisons skip more recent ZSL techniques in that the proposed IGSC formulation does not consider semantic vectors of unseen classes. However, the semantic vectors of both seen and unseen classes are often made available in the training stage of ZSL and most recent ZSL techniques do consider exploring such information. It would be useful if the proposed IGSC can be generalized to take account of all the available semantic information so that comprehensive comparisons to SOTA ZSL techniques can be carried out.

After author response:
My main concerns about this paper are technical novelty and weak experimental results. The authors did not make efforts to address the two aspects properly. For example, I have listed several issues in my comment 3 about the specific experimental results, but the authors did not try to address them at all. Their responses to Reviewer 4 about the transductive ZSL are not relevant to the weak experimental results of concern.  As most of my concerns are not resolved in the response, I see no evidence to upgrade my rating.

I appreciate that the authors have revised their responses. However, the added response still does not address the two specific questions (the reasons why AREN is not included in Table 3, and AREN+CS not in Table 4) in my comment 3. My final evaluation about the paper is on the negative side in that its technical novelty is moderate and the experimental results are not convincing.

---

> ### Author Response · Authors · 2020-11-23
> **Responses to AnonReviewer1**
>
> 1. Thank you for indicating the VQA work. Although a similar idea has been explored in solving the VQA problem, such a powerful idea has not been investigated for visual recognition, especially with the zero-shot learning setting. This technique is particularly useful to ZSL because the relationships of the visual evidences and the attributes---learned globally from seen classes---can help recognition involving unseen classes. Learning this correspondence is even suitable for solving image recognition because in VAQ one image usually has only a few questions while in our case one class has many images.
>
> 2. Thank you for giving the comment. We used the same math notations as those in the SP-AEN paper (Chen et al., CVPR 2018). We have fixed it in the paper.
>
> 3. AREN is the state-of-the-art embedding-based approach. It involves two branches of region attention mechanisms in the model design. We have corrected the acc_u value of SUN by AREN. Both approaches can benefit from the CS trick. If CS is not used, AREN achieved 25.5/52.1/26.7/16.4, while our results are 25.7/39.3/32.1/22.5. Except CUB, we outperformed AREN in all datasets.
>
> 4. Please refer to the response to AnonReviewer4.

---

### Official Review · AnonReviewer4 · 2020-10-29
**Good paper, easy to read with promising results**

**Rating:** 4
**Confidence:** 4

**Review:**

This paper proposes a visual-semantic embedding model useful for generalized zero-shot learning. The proposed model transforms an image into a label classifier, which is then used to predict the correct label in the semantic space.

The paper is well constructed and easy to read. It provides a good presentation of some related work and identifies the contributions as compared to existing approaches.

The experimental validation is performed on four popular public datasets and compares the performance to several state of the art approaches. The obtained performance shows similar/promising results as compared to the state of the art.

From my perspective, the paper is missing some experimental analysis/comparison to some recent methods that are inductive only to samples and also some methods that are transductive for unseen class prototypes and unlabeled unseen test instances
(for instance, papers mentioned in Section 4.3). First, that comparison will allow evaluating the performance of the proposed approach to more recent papers than the ones used in Section 4.2. Second, it seems that these methods, specifically the ones that are transductive for unseen class prototypes, achieve a much higher performance and it's important to evaluate the performance of the proposed method in that setting or to report on the performance loss when someone decides to use this approach in that specific setting (inductive to both unseen images and unseen semantic vectors).  A discussion that addresses the above questions/concerns could do it too.

Post-Rebuttal Evaluation [FINAL] I would like to thank the authors for their response and for updating their paper based on the reviewers feedback. Following these updates, I'm changing my recommendation to Rejection for the following 2 reasons: 1- the technical novelty of this paper is moderate and there's a significant gap in the model performance as compared to state of the art, 2- the authors failed to provide convincing answers to many of the reviewers concerns, including motivation for not using semantic embeddings during the training process, and not comparing their approach to  transductive ZSL ones which achieve a higher performance.

---

> ### Author Response · Authors · 2020-11-23
> **Response to AnonReviewer4**
>
> We agree that it is important to evaluate the performance of our method in the transductive setting. Based on our survey, there are two possible solutions: (1) integrating GAN-based methods for generating synthetic features of unseen classes; (2) manipulating semantic vectors for introducing a reconstruction loss to regularize the model. We have been experimenting with GAN-based methods (i.e. a simplified VAEGAN similar to f-VAEGAN-D2 proposed in CVPR 2019) but have not been successful for training a reasonable feature generator by far. For the second solution, because we do not apply any transformation to the semantic vectors (they are solely used to evaluate the learned label classifier), we found it non-trivial to construct a reconstruction loss.

---

### Official Review · AnonReviewer2 · 2020-10-29
**AnonReviewer2 [Finalized after authors' response]**

**Rating:** 3
**Confidence:** 5

**Review:**

*Summary*
The authors tackle the problem of zero-shot learning, that is, the recognition of classes and categories for which no visual data are available, but only semantic embedding, providing a description of the classes in terms of auxiliary textual descriptions. To this aim, authors propose a method dubbed  Image-Guided Semantic Classification in which a two-stream network (fed by either visual and semantic embedding) learns a compatibility function whose recognition performance is enhanced by means of calibrated stacking (Chao et al. 2016).

*Pros*
* The method is simple and easy to understand.


*Cons*
* The computational pipeline is not novel and largely inspired by methods such as the ones proposed in [Yang et al., A Unified Perspective on Multi-Domain and Multi-Task Learning, ICLR 2015] or [Liu et al. Generalized Zero-Shot Learning with Deep
Calibration Network, NeurIPS 2018] which are not even cited in the paper, unfortunately. Authors seem essential to add a calibration module to this kind of architectures: since the calibration module is inherited from prior works (Chao et al. 2016), I found the method quite incremental.

* Within the experimental comparison, GAN-based methods are not reported although being a mainstream class of state-of-the-art methods. I am referring to works such as CLSWGAN [Xian et al. Features Generating Networks for Zero-Shot Learning, CVPR 2018], f-VAEGAN-D2 [Xian et al. A Feature Generating Framework for Any-Shot Learning, CVPR 2019],  CADA-VAE [Schonfeld et al., Generalized zero- and few-shot learning via aligned variational autoencoders, CVPR 2019], DLFZRL [Tong et al., Hierarchical disentanglement of discriminative latent features for zero-shot learning, CVPR 2019] or tf-VAEGAN [Narayan et al. Latent Embedding Feedback and Discriminative Features for Zero-Shot Classification, ECCV 2020]. Authors justified this approach by reducing the methods in comparison within the ones which use unseen semantic embeddings only at test time. However, the knowledge of semantic embeddings also for the unseen classes is something which is necessary in a zero-shot recognition paradigm: the authors themselves take advantage of them during the nearest neighbor search. Thus, at this point, using the semantic embeddings for the unseen classes is therefore legit and I do not see any added value in constraining the experiments.

* The reported performance is highly suboptimal with respect to the state-of-the-art: invertible zero-shot recognition flows, recently proposed at ECCV 2020, greatly outperformed the proposed approach by margin: H=49.8 on aPY, H=54.8 on SUN, H=59.4 on CUB and H=68.0 on AWA2.

*Pre-Rebuttal Evaluation*
I regret to register a substantial overlap with prior methods, thus undermining the novelty impact of the present submission. On the experimental side, I found a sharply gapped performance which is highly inferior with respect to the state-of-the-art, caused by reducing the approaches included in the comparison on the basis of which methods exploit unseen semantic embeddings for training (this claim is not convincing in my opinion, unseen semantic embeddings are used in any cases, why not exploiting them for feature generation purposes?). For those reasons I am afraid to discourage the acceptance of the manuscript

*Post-Rebuttal Evaluation [FINAL]*
I would like to thank the authors for their response and for the updated version of the manuscript, I appreciate their efforts. Unfortunately, I still believe that the paper lacking about original contribution and I am not fully convinced by the authors' comments on the relationship with [Yang et al., A Unified Perspective on Multi-Domain and Multi-Task Learning, ICLR 2015] and [Liu et al. Generalized Zero-Shot Learning with Deep Calibration Network, NeurIPS 2018] which I still judge highly overlapping with the current methodology.
Furthermore, although authors clarified on the avoidance of using semantic embeddings for the training methodology, I do not see a sharp point in pursuing this approach given the high gap in performance with prior art (GAN-based).
For all these reasons, I regret to confirm my initial rejection score.

---

> ### Author Response · Authors · 2020-11-23
> **Responses to AnonReviewer2**
>
> 1. Thank you for indicating the methods [ICLR 2015 and NeurIPS 2018]. Both methods project visual and semantic representations to a latent space and perform classification by the nearest prototype classifier (NPC), which in our view apply the conventional zero-shot learning paradigm as illustrated in Fig. 1 (a) and (b) in our paper. In the ICLR 2015 paper, $P$ and $Q$ are static during inference. Similarly, the prediction function $f$ in the NeurIPS 2018 paper is fixed once trained. We propose a completely different approach, in that the model (label classifier) construction is adaptive to the input image during test-time. We will cite the papers in our papers.
> Indeed, the calibration module is not the focus of our paper. We directly use this trick from Chao et al. 2016, as it additionally improves the performance.
> 2. Thank you for referring GAN-based methods. The settings are different in that GAN-based methods require the semantic embeddings for the unseen classes but we do not. We are confused about the sentence about "the authors themselves take advantage of them" because, we neither use them (the semantic embeddings for the unseen classes) nor the nearest neighbor search (i.e. we did not perform label classification via nearest neighbor search). We agree with you and other reviewers that we should not restrict ourselves to the instance- and class-inductive setting. We agree that GAN-based methods provide a very strong data augmentation scheme and therefore should significant benefit our method for boosting the recognition performance.
> 3. To achieve state-of-the-art performance, one method must have a powerful feature generating method (like GAN-based methods) and an effective classifier (like the focus of our paper). Most GAN-based methods focus on the former and use linear softmax classifiers, which we believe these STOA methods can be improved by using the proposed, adaptive classification approach. We have added a paragraph (the last paragraph in the experimental section) that compares our method to GAN-based methods.

---

### Author Response · Authors · 2020-11-25
**Summary of changes**

We would like to thank all reviewers for their valuable and constructive comments, which help us improve the paper.
1. We have added a paragraph in the experimental section discussing the comparison to GAN-based methods.
2. We have added subsection 4.4 discussing the model flexibility and visualizing the generated classifier weights.
3. We have fixed all errors mentioned in the comments.

Below please find our response to each comment.

---

### Decision · Program_Chairs · 2021-01-07
**Final Decision**

**Decision:**

Reject

**Comment:**

This paper presents work on zero-shot learning.  The reviewers appreciated the simplicity of the method and its clear exposition.  However, concerns were raised over novelty, motivation, and empirical validation.  After reading the authors' response, the reviewers remained of the opinion that these concerns have not yet been addressed sufficiently.  Based on these points, the paper is not yet ready for publication.